**Data Availability Statement:** Data cannot be shared publicly because American Indian youth are a vulnerable population and the study does not

## RESEARCH ARTICLE

# Psychometric evaluation of protective measures in Native STAND: A multi-site cross-sectional study of American Indian Alaska Native high school students

**Allyson Kelley**[1], **Thomas McCoy**[2], **Megan Skye**[3], **Michelle Singer**[4], **Stephanie Craig Rushing**[4]*, **Tamara Perkins**[5], **Caitlin Donald**[3], **Kavita Rajani**[3], **Brittany Morgan**[6], **Kelley Milligan**[1], **Tosha Zaback**[3], **William Lambert**[3]

**1** Allyson Kelley and Associates, Sisters, Oregon, United States of America, **2** School of Nursing, University of North Carolina Greensboro, Greensboro, North Carolina, United States of America, **3** Obstetrics and Gynecology, Section of Family Planning, Oregon Health and Science University, Portland, Oregon, United States of America, **4** Northwest Portland Area Indian Health Board, Portland, Oregon, United States of America, **5** NPC Research, Portland, Oregon, United States of America, **6** Department of Public Health Sciences, University of California Davis, Davis, California, United States of America

* scraig@npaihb.org

## Abstract

American Indian and Alaska Native (AI/AN) youth are strong in culture and rich in heritage and experience unique strengths and challenges throughout adolescence. Documenting conditions that protect against risk factors associated with poor health outcomes are needed. We explored scales that measure self-esteem, culture, social support, and community from a sample of 1,456 youth involved in Native STAND, a culturally-relevant evidence-based sexual health intervention. We established content validity by reviewing existing literature and community feedback. Construct validity was examined using factor analysis. The final self-esteem model included seven items, factor loadings ranged from 0.47 to 0.63 for positive self-esteem and 0.77 to 0.81 for negative self-esteem. The final culture model included three items, factor loadings 0.73 to 0.89. The social support scale included four items, factor loadings ranged from 0.86 to 0.87 for family social support and 0.75 to 0.77 for friends social support. The community and community safety scale included three items; factor loadings ranged from 0.52 to 0.82. Coefficient alphas for scales ranged from $\alpha = 0.63$ to $\alpha = 0.86$. This study validated scales in a national sample of AI/AN youth–psychometric scales provide an essential tool for documenting the needs and strengths of AI/AN youth.

## Introduction

The 577+ American Indian and Alaska Native (AI/AN) tribes and communities throughout the U.S. are diverse in their language, culture, traditions, spiritual practices, and heritage. This diversity must be considered when developing and utilizing measures to assess the effectiveness of health and educational programs, interventions, and policies. AI/AN youth under 18 years old make up about 29% of the AI/AN population–too many experience poor mental

allow for data sharing. Data are available by contacting Dr. Craig and going through the appropriate IRBs and Ethics Committees for researchers who meet the criteria for access to confidential data. The Portland Area Indian Health Service IRB contact is Rena Macy, Co-Chair, Portland Area IHS IRB Portland Area IHS 1414 NW Northrup St Suite 800 Portland, OR 97209 Phone: 503-414- 5540 It should be noted that all shareable data are within the paper and its Supporting Information files. Data sharing is not appropriate for this small sample of AI/AN youth. Data could potentially be identified and AI/AN youth populations are considered vulnerable populations due to the historical and unethical research practices conducted by universities and the US government on AI/AN populations. If a reader would like access to the data used in this study they would need to submit a request to Dr. Stephanie Craig Rushing, the corresponding author and she would present this information to the tribe and IRBs involved in the Native STAND study.

**Funding:** This work was supported in part by the award number 5 U48DP005006-05 from the Centers for Disease Control and Prevention, Cooperative Agreement. The content is solely the responsibility of the authors and does not necessarily represent the official views of the Centers for Disease Control. There was no additional external funding received for this study. The funders had no role in study design, data collection and analysis, decision to publish, or preparation of the manuscript.

**Competing interests:** The authors have declared that no competing interests exist.

health outcomes resulting from disproportionate exposure to historical trauma and other social, structural, political, and environmental factors [1]. As a result, AI/AN youth are placed at greater risk for suicide, poor mental health, depression, and substance misuse [2].

Culturally-adapted evidence-based interventions (EBIs) are emerging as an essential tool for building health equity in AI/AN populations [3]. Assessing the validity of instruments that document the characteristics of program participants and program impact is essential. Without valid instrumentation, it is nearly impossible to know if an intervention worked, for whom, and in what context. For decades, researchers, policymakers, educators, and interventionists have been challenged with data collection in AI/AN populations, including restricted data access, limited capacity to analyze data, difficulty finding comparison groups, limited quality data, small sample sizes, lack of national AI/AN samples, limited comparison groups, lack of cultural validity, and varying levels of programmatic capacity [4–5]. Researchers often use assessment instruments designed for the general population in AI/AN contexts–often, these lack cultural context or community feedback [5]. Developing valid instruments with AI/AN communities requires an understanding of differences in views about what determines health and wellness and the unique community and cultural practices that protect youth. The language and terms used to define health outcomes in mainstream survey tools may be based on western medialized constructs that have little relevance in AI/AN community settings [5]. Researchers working with AI/AN communities to evaluate health programs and interventions need to reliably measure risk and protective factors among program participants. In this review, protective factors refer to conditions or variables that may increase positive health outcomes in AI/AN youth.

Research suggests that self-esteem, social support, community connections, and culture may be protective [6]. Self-esteem has been associated with school success, internal locus of control, perceptions of competence, social support, and coping skills [7]. Peer and family social support has been inversely correlated with AI/AN youth substance misuse [8]. Family caring has been associated with positive AI/AN youth mental health [9]. Cultural connectedness has been associated with academic success and resilience [10]. One study found that AI/AN youth with an interest in their culture were less likely to display violent behaviors [11]. Stable community conditions may foster healthy connections and safety [8, 12], which may be protective against substance misuse and suicide ideation. Such research demonstrates forward progress in understanding what determines health in AI/AN youth. Still, more work is needed to capture the strengths of AI/AN youth and communities through the development of valid survey measures.

Attempts to measure protective factors such as self-esteem, social support, community, and culture are evident in standardized assessments, but few have been developed with AI/AN youth in mind. First, Rosenberg's self-esteem scale has been used to assess participant self-esteem characteristics in the general population [13]. Second, Zimet and colleagues developed a Multidimensional Perceived Social Support scale [14]–which is widely used in the health programming milieu. Unlike social support and self-esteem, where standard mental health assessments are used or adapted, assessing aspects of culture and ethnic identity requires a different approach. Snowshoe and colleagues developed the Cultural Connectedness Short Scale (CCSS) to measure culture in First Nations youth [15]. The CCSS is meant to be adapted by First Nations and Native communities based on their unique culture, traditions, and context. Importantly, culture and community depend on values, history, traditions, customs, sense of belonging, and membership–the construct of culture is often defined and conceptualized with a specific community in mind [16].

Much has been done to develop and evaluate culturally-relevant health interventions that build on the strengths and resilience of AI/AN youth. Less has been done to assess the

effectiveness of these interventions using validated survey tools appropriate for the AI/AN community and population. The objective of this study was to validate the measurement properties of the Native Students Together Against Negative Decisions (Native STAND) survey instrument among AI/AN high school students.

## About Native STAND

Native STAND is a comprehensive sexual health curriculum for Native high school students that supports healthy decision-making through interactive discussions and activities that promote self-esteem, goals and values, team building, negotiation and refusal skills, and effective communication. The 90-minute lessons contain stories from Tribal communities that ground learning in cultural teachings. Results from previous Native STAND evaluations demonstrate it to be an effective approach for addressing healthy relationships, STDs, and teen pregnancy [17]. The original STAND intervention was designed and evaluated among rural youth in the southern United States and found to promote condom self-efficacy, STI risk behavior knowledge, and conversations with peers about other sexual health topics among participating students. Previous evaluations of the Native STAND curriculum conducted from 2010 to 2012 with a sample of 90 students reported positive results, with increases in STD/HIV knowledge, reproductive health, and healthy relationships [17]. The original survey tool was developed by the Centers for Disease Control and focused heavily on sexual health knowledge, attitudes, and behavior. For the present study, we cut back the survey tool, updated the demographic questions, and expanded the protective measures. No prior studies have established the reliability and validity of the tool's protective measures.

## Methods

The Native STAND D&I research study was a collaboration between [REMOVED FOR BLINDED REVIEW] and 48 Tribes and Native-serving organizations located across the U.S. The study protocol was reviewed and approved by OHSU (IRB00000734) and the Portland Area Indian Health Service Institutional Review Board (659942).

### Sample

Students were located in 17 states, including Alaska. Most of the sites were from rural communities (82.7%) and were from the western part of the United States. The youth who signed up to participate in Native STAND filled out the Native STAND survey prior to participation, from September 2015 to March 2019.

### Procedure

In 2014 the study team began revising the Native STAND survey tool, which was adapted from the Native Youth Survey and other questionnaires [17]. During the first round of site recruitment, the study team asked educators to review and discuss the survey instrument during an in-person orientation/training. Changes to the initial survey tool included reducing the number of self-esteem items, eliminating positive outlook, morals, and values, adaptability, and some measures on cultural pride and identity, to focus the instrument on the most important measures and minimize survey fatigue [18]. Facilitator feedback and literature reviews were used to establish the content validity of the survey measures. The final survey tool was utilized with three cohorts of students at 48 sites and assessed a broad range of health knowledge, attitudes, beliefs, intentions, behaviors, and skills relating to physical, sexual, mental, and psychosocial health. This study focuses on four protective measures included in the updated Native STAND survey tool: self-esteem, culture, social support, and community safety.

Parental consent and youth assent were obtained prior to data collection. To ensure the confidentiality of survey responses, each youth received a paper survey labeled with a unique study ID and a manila envelope. After the youth completed their survey, they sealed them in the envelope and returned them to the educator on site. Educators then mailed their class's surveys to the Native STAND project office for data entry and cleaning. Once received, researchers would immediately scan the survey for any safety concerns. If a safety concern was noted, educators would be notified that a safety concern was noted among their students so they could address it with safety protocols they had in place within their institution.

## Measures

**Self-esteem.** Participants were asked their level of agreement with seven statements based on questions selected from the Rosenberg Self-esteem scale [13] by the study team and reviewed with participating health educators (e.g., I take a positive attitude toward myself). The team examined the VOICES survey [7] and the original Rosenberg Self-esteem scale, including ten items, five positively worded and five negatively worded. The study team reduced the number of items on the scale from 10 to seven and omitted one negatively worded item [7, 13,18]. Previous research reports the potential low reliability of negatively worded items in youth and culturally-diverse populations [7]. Response options were based on a Likert-style scale where 1 = Strongly Disagree to 5 = Strongly Agree.

**Culture.** Participants were asked to rate their level of agreement with three statements based on culture questions identified by the study team and reviewed with participating health educators (e.g., I believe that I have many strengths because I am Native American).

Social support. Participants were asked to rate their level of agreement with four statements based on questions identified by the study team and reviewed with participating health educators (e.g., I can talk about my problems with my friends).

**Community and community safety.** Participants were asked to rate their level of agreement with three statements based on questions identified by the study team and reviewed with participating health educators (e.g., I feel safe in my community or neighborhood).

[See S1 File]

## Data analysis

Item analyses were first performed for descriptive statistics of item ratings (mean (*M*), standard deviation (*SD*)) and sample size of youth with complete responses. Inter-item correlations among responses as well as corrected item-total correlations were estimated. Exploratory factor analysis (EFA) was performed using principal axis factoring exaction with an oblique promax rotation. The number of factors extracted was based on parallel analysis [19]. An item was considered to load onto a factor if its factor loading was at least 0.40. Non-loading items and items loading on more than one factor were removed and the EFA re-ran. Internal consistency reliability was estimated for each factor using Cronbach's alpha ($\alpha$). All analyses were performed in SPSS v26 (IBM Corp., Armonk, NY) and M*plus* v8.3 [20]. A two-sided *p*-value $< 0.05$ was considered statistically significant.

## Results

### Sample characteristics

Our sample included 1,456 students, with a median age of 15, and were 50.2% female, 45.7% male, and 1.2% transgender (Table 1). Students represented 48 sites from throughout the US, most (84.9%) were American Indian.

**Table 1. Characteristics of Native STAND participants, 2015–2019 (*N* = 1,456).**

| Participant Characteristic | *n* (%) |
|---|---|
| Age (Median [IQR]) | 15 [2] |
| Gender | |
| Female | 731 (50.2) |
| Male | 665 (45.7) |
| Transgender | 17 (1.2) |
| missing | 43 (3.0) |
| Race/Ethnicity | |
| American Indian/Alaska Native | 1,236 (84.9) |
| Other (non-White) | 179 (12.2) |
| Sexual Orientation | |
| Straight/Heterosexual | 1,045 (71.8) |
| LGBTQ2S+ | 129 (8.9) |
| Unsure/Don't Know | 188 (12.9) |
| missing | 94 (6.5) |
| Geographical distribution | |
| Oregon | 166 (11.4) |
| Arizona | 394 (27.1) |
| New Mexico | 401 (27.5) |
| Other | 495 (34.0) |

*Note. IQR = Interquartile range.

## Descriptive results

Table 2 displays descriptive results of the Native STAND student participants. Mean scores ranged from 2.5 to 4.4.

**Self-esteem.** Item ratings varied from 3.4 for "calling" to 4.3 for "best" on average (*SD* ranged from 0.72 to 0.95) with a sample size of 1,373 youth with complete item responses. Inter-item correlations ranged from 0.213 to 0.329 (average = 0.278) and corrected item-total correlations ranged from 0.361 to 0.628. Two factors of self-esteem resulted based on parallel analysis, which explained 53.9% of the variance in the seven self-esteem items (see Fig 1). The two reverse scored (negatively worded) items loaded on their own factor (Negative Self-esteem) while the remaining five not reverse scored loaded onto their own factor (Positive Self-esteem), with a correlation between them of 0.442. Factor loadings ranged from 0.47 to 0.63 for Positive Self-esteem (Cronbach's $\alpha$ = 0.655) and 0.77 to 0.81 for Negative Self-esteem ($\alpha$ = 0.771).

**Culture.** Item ratings varied from 3.8 for "history" and "native strengths" to 4.2 for "native identity" on average (all *SD* = 1.0) with a sample size of 1,390 youth with complete item responses. Inter-item correlations ranged from 0.565 to 0.692 (average = 0.636) and corrected item-total correlations ranged from 0.661 to 0.759. One factor of AI/AN Culture resulted based on parallel analysis which explained 75.8% of the variance in the three culture items (Fig 2). Factor loadings ranged from 0.73 to 0.89, and internal consistency was high ($\alpha$ = 0.840).

**Social support.** Item ratings varied from 3.6 for "talk problems" to 4.0 for "support" on average (*SD* ranged from 0.94 to 1.1) with a sample size of 1,422 youth with complete item responses. Inter-item correlations ranged from 0.286 to 0.759 (average = 0.439) and corrected item-total correlations ranged from 0.588 to 0.758. Two factors of social support resulted based on parallel analysis, which explained 83.7% of the variance in the four social support items (Fig 3). The two "family" related items loaded on their own factor (Family social support) while the two "friends" related items loaded onto their own factor (Friends social

**Table 2. Item location within constructs of Native STAND participants, 2015–2019.**

| Construct/Item | M (SD) | $r_{\text{item-total}}$ |
|---|---|---|
| **Self-Esteem (n = 1,373)[a]** | | |
| I smile and laugh a lot | 4.1 (0.86) | .426 |
| I adjust well to new situations and challenges | 3.7 (0.82) | .411 |
| I try to do my best | 4.3 (0.72) | .397 |
| I am optimistic about my future | 3.9 (0.86) | .456 |
| I have a sense of what life is calling me to do | 3.4 (0.95) | .361 |
| Sometimes I think I am no good at all (RV)[a] | 3.0 (1.16) | .628 |
| I feel that I am a failure (RV) [a] | 2.5 (1.13) | .628 |
| **Culture (n = 1,390) [a]** | | |
| Being Native American is a major part of my identity | 4.2 (1.02) | .759 |
| I believe that I have many strengths because I am Native American | 3.8 (1.02) | .661 |
| I have spent more time trying to find out more about the history, traditions, and customs of Native people | 3.8 (1.04) | .691 |
| **Social Support (n = 1,422) [a]** | | |
| If I had a personal problem, I could ask someone in my family for help | 3.7 (1.15) | .758 |
| Share thoughts/feelings family | 3.7 (1.12) | .758 |
| I have friends who support me | 4.0 (0.94) | .588 |
| I can talk about my problems with my friends | 3.6 (1.13) | .588 |
| **Community (n = 1,416) [a]** | | |
| I feel safe in my community or neighborhood | 3.9 (0.90) | .538 |
| If I had to move, I would miss the community I now live in | 3.9 (1.14) | .432 |
| I feel safe at home | 4.4 (0.77) | .405 |

[a] Individual variable denominators differ depending on missingness. Anchor ratings of 1 = Strongly Disagree to 5 = Strongly Agree.

[b] Reverse Coded (RV). M = mean, SD = standard deviation, $r_{\text{item-total}}$ = corrected item-total correlation.

support), with a correlation between them of 0.475. Factor loadings ranged from 0.865 to 0.874 for Family social support ($\alpha$ = 0.732) and 0.759 to 0.770 for Friends social support ($\alpha$ = 0.862).

**Community and community safety.** Ratings varied from 3.9 for "would miss" and "feel safe" to 4.4 for "home" on average (SD range 0.77 to 1.1) with a sample size of 1,416 youth with complete item responses. Inter-item correlations ranged from 0.279 to 0.439 (average = 0.382) and corrected item-total correlations ranged from 0.405 to 0.538. One factor of Community resulted based on parallel analysis, which explained 59.0% of the variance in the three community items (Fig 4). Factor loadings ranged from 0.52 to 0.82, and internal consistency was high ($\alpha$ = 0.635).

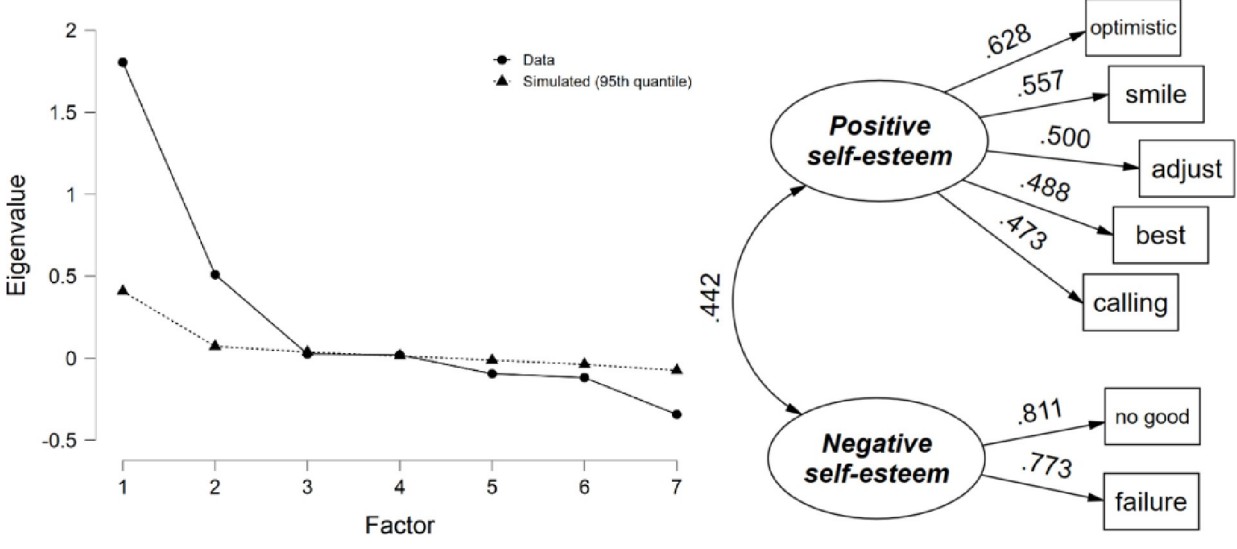

**Fig 1. Self-esteem parallel analysis scree plot and path diagram.**

## Discussion

The Native STAND survey scales reported here demonstrate the validity and reliability of strength-based survey measures in AI/AN youth participating in the Native STAND multi-site cross sectional study. The participatory process used to develop the survey tool and implement the intervention with diverse AI/AN communities added to the content validity of survey questions and provides a framework for other communities and researchers moving forward.

Previous research on the psychometric properties of strength-based scales have not been specific to AI/AN populations or the Native STAND intervention. Findings fill an essential gap in the literature–documenting the validity and reliability of strength-based measures in a national sample of AI/AN youth. Results presented here are a starting point for researchers, communities, and programs as they continue developing surveys with solid psychometric priorities for documenting the effectiveness of various programs, policies, and interventions.

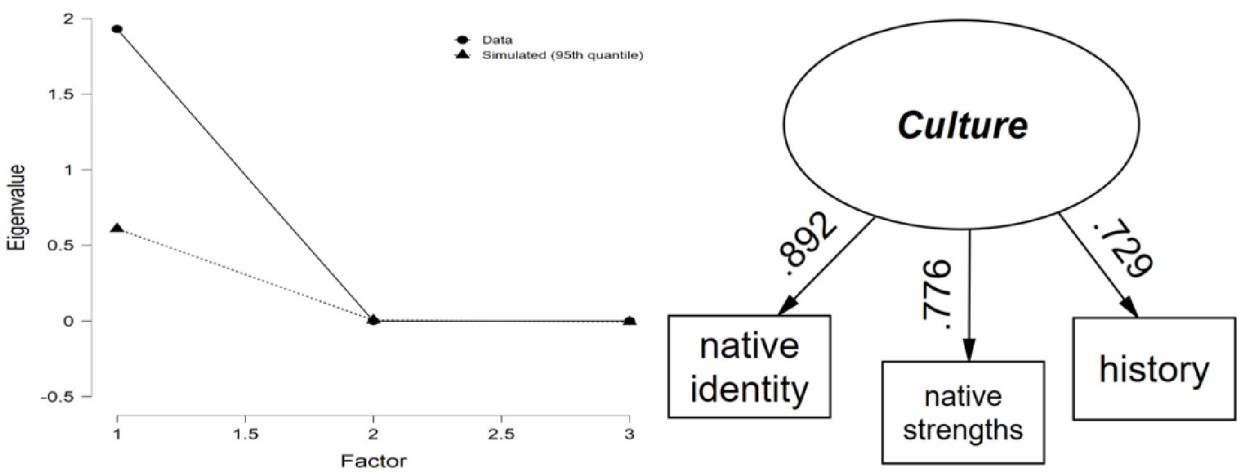

**Fig 2. Culture parallel analysis scree plot and path diagram.**

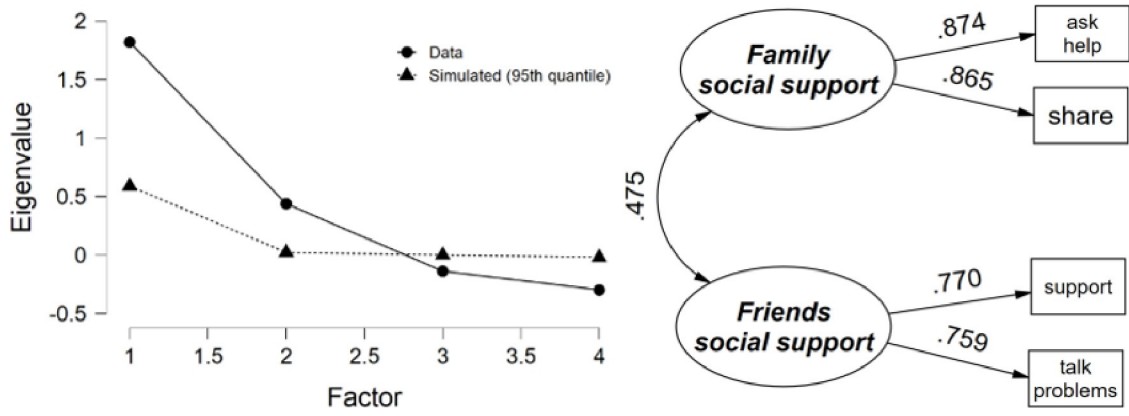

**Fig 3. Social support parallel analysis scree plot and path diagram.**

Results from the factor analysis and internal consistency provide guidance on creating and using sum scores for each factor/subscale.

Future efforts can build on this preliminary work, measuring protective factors in AI/AN youth that promote health and healthy decision-making [21]. For example, researchers may consider more comprehensive survey measures that define and operationalize culture and Native American identity, for example, creating culture-specific questions about involvement in history, traditions, and customs of a specific tribal group. At the individual level, future protective and strength-based surveys may include questions about healthy coping and problem-solving skills, emotional self-regulation, academic achievement, and positive physical development [7]. At the community level, additional questions could expand on the role of school, neighborhood, and community while documenting the presence of mentors, support, positive norms, behavior expectations, and safety [6]. Family strengths and protective factors are also essential, and documenting how the family provides structure, limits, rules, monitoring, predictability, supportive relationships, and behavior expectations is warranted. The Native STAND team encourages researchers, policymakers, programs, communities, clinicians, and educators to consider what is protective and what questions are essential to a program or study. Meeting with communities and members of the focus population can help define what

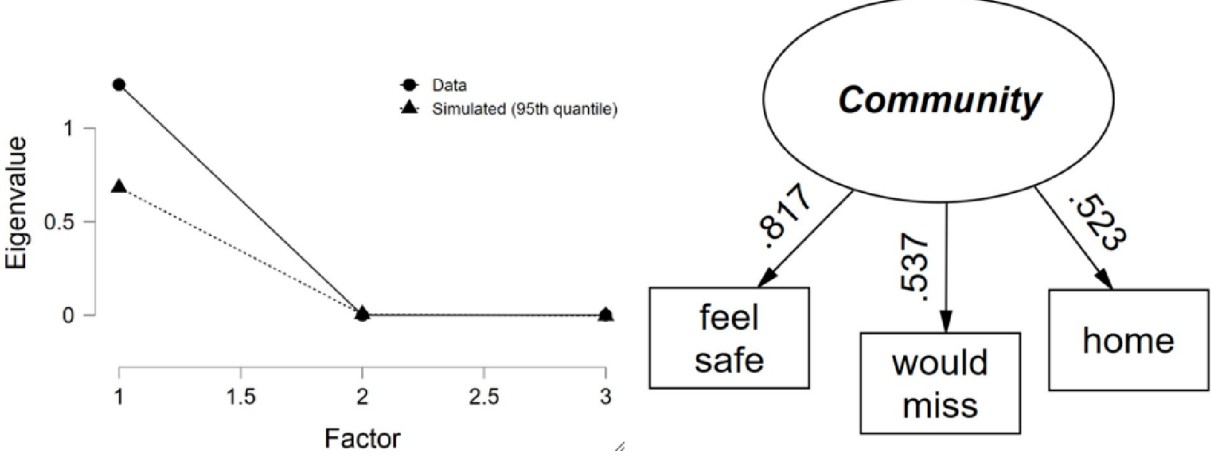

**Fig 4. Community parallel analysis scree plot and path diagram.**

wellness, health, culture, and community looks like and create surveys that build on these strengths and resources [5]. Although the Native STAND survey does not capture every strength-based measure possible, it is a starting point for documenting the strengths and needs of AI/AN youth throughout the life course.

## Limitations

Although the contributions of this study are clear, there are a few limitations that should be noted. First, information collected from Native STAND youth is based on self-report responses; this could result in social desirability bias. The Native STAND team provided confidential areas for data collection and ensured anonymity throughout the intervention. Secondly, construct validity was not assessed in this study, and the team was not able to compare responses with actual behaviors. Last, although Native STAND was conducted with a national sample of AI/AN high school youth, these youth do not represent all populations, communities, or cultural perspectives of the 577+ tribal nations in the U.S. It is possible that the reliability and validity of these scales may be different with different populations.

## Conclusion

This study validated strength-based scales in a national sample of AI/AN youth involved in the Native STAND study. Findings support the use of strength-based measures and community participation in the research process as opposed to deficit-based measures and problem behaviors. Protective and strength-based survey measures are beneficial to individuals and the community as a whole because they demonstrate what is positive in a youth's life and what promotes health. This knowledge is often transferrable at the individual, family, and community levels.

## Supporting information

**S1 File. This is the S1 File Native STAND survey measures.**
(DOCX)

## Acknowledgments

The authors acknowledge the incredible efforts made by youth, educators, Tribes, and schools who completed the Native STAND survey. We appreciate your dedication to wellness and the Native STAND curriculum.

## Author Contributions

**Conceptualization:** Allyson Kelley, Michelle Singer, Stephanie Craig Rushing.

**Data curation:** Megan Skye.

**Formal analysis:** Thomas McCoy.

**Investigation:** Stephanie Craig Rushing, William Lambert.

**Methodology:** Thomas McCoy.

**Project administration:** Stephanie Craig Rushing, Caitlin Donald, Kavita Rajani, Brittany Morgan, Tosha Zaback.

**Writing – original draft:** Allyson Kelley.

**Writing – review & editing:** Allyson Kelley, Thomas McCoy, Tamara Perkins, Kelley Milligan.

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
