## [Decision Letter · Decision Letter 0]

24 Jan 2022

PONE-D-21-24780Psychometric evaluation of measures in

the Native STAND study of American Indian Alaska Native YouthPLOS ONE

Dear Dr. Kelley,

Thank you for submitting your manuscript to PLOS ONE. After careful consideration, we feel that it has merit but does not fully meet PLOS ONE’s publication criteria as it currently stands. Therefore, we invite you to submit a revised version of the manuscript that addresses the points raised during the review process.

We look forward to receiving your revised manuscript.

Kind regards,

Richard Huan XU

Academic Editor

PLOS ONE

https://journals.plos.org/plosone/s/file?id=ba62/PLOSOne_formatting_sample_title_authors_affiliations.pdf”.

“This work was supported in part by the Centers for Disease Control and Prevention, Cooperative Agreement Number [5 U48DP005006-05].  “

“No authors have competing interests.”

Reviewers' comments:

Reviewer's Responses to Questions

**Comments to the Author**

1. Is the manuscript technically sound, and do the data support the conclusions?

Reviewer #1: Partly

Reviewer #2: Partly

2. Has the statistical analysis been performed appropriately and rigorously? 

Reviewer #1: Yes

Reviewer #2: Yes

3. Have the authors made all data underlying the findings in their manuscript fully available?

Reviewer #1: No

Reviewer #2: No

4. Is the manuscript presented in an intelligible fashion and written in standard English?

Reviewer #1: Yes

Reviewer #2: Yes

5. Review Comments to the Author

Reviewer #1: Thanks for the opportunity to let me review the manuscript. This study conducted a psychometric evaluation for a survey tool used in Native STAND study for American Indian/Alaska Native Youth. After review, I found some weakness in its results and discussion. Detailed comments can be found below.

1. Line 148-150, the authors stated that they added questions on multiple domains in the survey tool, why only focus on the four domains in the study? Inclusion of the other new domains could give a full picture of validity and reliability of the new tool, please consider to add them, or explain why these four domains are particular important.

2. Could the authors also report the average inter-item correlations (in addition to the range)? The details of the pairwise inter-item correlations and the corrected item-total correlations can be put in a supplementary file for audience who are interested to know the details of the tool. For the “α” in the results, please specify that they are Cronbach α (if not, please define this indicator).

3. From the results, I can only tell the scales have a reasonable internal consistency. How about the other aspects of the validity and reliability. For example, did the authors test the criterion validity of the survey tool?

4. Did the authors test the test-retest reliability of this tool, for all or a subset of the participants? It could evaluate the stability of the measurement through time, which is an important component of its psychometric property.

5. What are the implications of the reported EFA outcomes in implementation of this tool and future studies, or how can people make use of the EFA outcomes in practice? Is it possible to generate a composite score for each of the domains or subscales using the EFA outcomes? Please consider to add this into the discussion.

6. In the discussion, the authors indicated that “The participatory process used to develop the survey tool and implement the intervention with diverse AI/AN communities provides a framework for other communities and researchers moving forward”; therefore, could the authors describe the process to develop the survey tool in details, particularly how the content validity was assessed and improved?

7. How were the reliability and validity of the survey tool in AI/AN youth compared with general population, especially the self-esteem that is included in both old and new version of the tool? Please describe it in the discussion.

8. How would the authors comment on the generalizability/external validity of the updated survey tool among the youth in other ethnic minority groups?

Reviewer #2: Thanks for inviting me to review this interesting manuscript.

In general, the authors had done tremendous works to obtain the data. Below I provide comments concerns which, I think, can be clarified by the authors and are necessary for the publication of the paper.

Q1

Title

1. In the title, I think the use of “youth” may not appropriate, as sample of this study focused merely on high school students with a median age of 15 years.

2. “Psychometric evaluation” means the manuscript must present specific results of the survey, and analyze their differences. But, the “objective” of this study (line 112-114, page 5) was to validate the measurement properties of …. And on the other hand, as a research of reliability and validation, I don't think this study can reach that conclusion (more details given in Q3), so it would be more appropriate to call it a pilot study or a cross-sectional study.

Q2

The presentation of the manuscript is not rigorous enough, and some small mistakes are obvious.

1. As far as known from the manuscript, sample number of this research was 1456 exactly, but, line 44-45 in page 1, the author why said “more than”?

2. Page 2, line 48, authors represented in the manuscript that the factor of negative self-esteem was two-item, so, factor loading of “0.65” was redundant.

3. I doubt about “resilience” you have mentioned in the keywords list, there is less content related to it.

4. Page 9, line 204, item ratings here were all included in the factor of “positive self-esteem”, it should be clearly defined and stated.

5. Page 9, line 213, items of “history” and “native strengths” had the same rating of 3.8.

Q3

Some of the statistical procedures were not clearly explained or the methods were incorrectly used. Because of this issue, the conclusion that the measurement is a reliable and valid tool should be reconsidered or "tone-downed".

1. As presented in figure 1. and figure 3. , parallel analysis of “self-esteem” and “social support” may indicate one more factor for each scale; these were also confirmed by the unsatisfactory total variance explanation (e.g., 53.9% of self-esteem and 59% of community of community safety). Please clarify.

2. Page 6, Line 138-139, participants of this study were high school students, they could not represent “youth”, and we know that, there are fifty states in the U.S., how these 16 states sampled, it may need more clarification.

3. Page 8, line 189-190, does any item was removed based on the EFA, and its procedure, I didn’t see it.

4. Page 6, Line 139- 140: “Most of the sites were from rural communities and were from the western part of the United States.” Maybe the geographical distribution of the samples could be more elaborated in Table 1, for clarification.

5. page 6, line 140, “a baseline survey” was done prior to participation, however, there is no description for the baseline. Please elaborate.

6. Line 148-150, the study team added questions about many factors, but the Manu did not show their results, and if not, why you mention them here. Please clarify.

7. Table 1: The sample consisted of 1140 AI/AN, and 150 others. Regarding Race/Ethnicity data, if 1140+150=1290 students were included in this study, which does not equal No.= 87.8% and No.=12.2%? Please explain.

Q5

Part of “measures”

1. Questions generating methods of the four scale and the scoring rules were identical, I think, there is no need to repeat the same description four times.

2. Line 182, where is the “supplement 1.” I didn’t see it.

Q6

“Conclusion” of the study was just derived a “start point”, and very little valuable information can be drawn from the current statement for fellow researchers.

6. PLOS authors have the option to publish the peer review history of their article (what does this mean?). If published, this will include your full peer review and any attached files.

Reviewer #1: No

Reviewer #2: **Yes: **Ling-ming Zhou

---

## [Author Response · Author response to Decision Letter 0]

29 Mar 2022

March 9, 2022

Dear Editor,

We are pleased to resubmit our manuscript for consideration in PLOS One. Our responses to reviewer comments are located in the table below.

Please feel free to contact me if you have additional questions. 

Allyson Kelley, DrPH

kelleyallyson@gmail.com

Reviewer 1 Response 

Line 148-150, the authors stated that they added questions on multiple domains in the survey tool, why only focus on the four domains in the study? Inclusion of the other new domains could give a full picture of validity and reliability of the new tool, please consider to add them, or explain why these four domains are particular important.

 Consistent with AIAN community protocols, this study focused on survey domains that were positive, strengths-based and protective. The inclusion of other domains beyond self-esteem, culture, social support, and community safety was beyond the study realm. The Native STAND team has published other research on all domains included in the Native STAND comprehensive sex education curriculum. These domains are beyond the scope of the present manuscript. 

Could the authors also report the average inter-item correlations (in addition to the range)? The details of the pairwise inter-item correlations and the corrected item-total correlations can be put in a supplementary file for audience who are interested to know the details of the tool. For the “α” in the results, please specify that they are Cronbach α (if not, please define this indicator). We have added the average inter-item correlations in the text when reporting the range. 

We also have already reported the range of the corrected item-total correlations so we decline to report an additional supplementary file. 

We have defined Cronbach’s α at first use.

From the results, I can only tell the scales have a reasonable internal consistency. How about the other aspects of the validity and reliability. For example, did the authors test the criterion validity of the survey tool?

 We did assess criterion validity of the scales with other studies. For example, the Native STAND efficacy study (Skye, 2021) reported on study outcomes related to sexual health intervention constructs and strength-based measures here, for example family social support. In the 2021 article, authors utilized this tool at baseline and follow-up and found some slight increases. 

Did the authors test the test-retest reliability of this tool, for all or a subset of the participants? It could evaluate the stability of the measurement through time, which is an important component of its psychometric property. No, this was not feasible due to the manner in which survey data were collected. In the future the team plans to test the test-retest reliability of the tool with other participants to assess stability of measurement. 

What are the implications of the reported EFA outcomes in implementation of this tool and future studies, or how can people make use of the EFA outcomes in practice? Is it possible to generate a composite score for each of the domains or subscales using the EFA outcomes? Please consider adding this into the discussion. EFA provides evidence of the number of factors, their loadings (i.e., measures of item discrimination), etc. It is a standard practice to create sum scores (or weighted sum scores) from the items that load of each factor. The implications for readers are that they can use subscale sum scores to analyse those factors in future research. We have added language in the Discussion around.

In the discussion, the authors indicated that “The participatory process used to develop the survey tool and implement the intervention with diverse AI/AN communities provides a framework for other communities and researchers moving forward;” therefore, could the authors describe the process to develop the survey tool in detail, particularly how the content validity was assessed and improved? We added a description of the process as it relates to content validity in the section before measures. Thank you. 

How were the reliability and validity of the survey tool in AI/AN youth compared with general population, especially the self-esteem that is included in both old and new version of the tool? Please describe it in the discussion. We addressed this comment and used the Rosenberg's self-esteem scale with AIAN youth as an example. We are not sure what you mean about the old and new version of the tool. Our team reduced the number of items in the self-esteem scale from 10 to 7. The original RSE included five positive and five negative items. Previous research reports that the negative or reverse coded (RV) items were not as reliable as the positively coded items. In the original scale a score of 30 to 40 indicates high self-esteem and internal consistency and test-retest correlations were good, Cronbach's alpha is 0.85 and 0.88. We expanded on this in the discussion and added a citation from the Voices of Indian Teens research study with AIAN youth which also utilized a modified six item RSE in the Native VOICES study and alphas ranged from .79 to .84 

Self-esteem (Likert 1=Strongly Disagree-5=Strongly Agree)

1. I smile and laugh a lot 

2. I adjust well to new situations and challenges 

3. I try to do my best

4. I am optimistic about my future 

5. I have a sense of what life is calling me to do 

6. Sometimes I think I am no good at all (RV)

7. I feel that I am a failure (RV)

How would the authors comment on the generalizability/external validity of the updated survey tool among the youth in other ethnic minority groups? While we cannot comment on the generalizability and external validity of the updated survey tool with other ethnic minority groups, we can say that the survey tool is applicable to other AIAN youth, communities, settings and interventions. Within AIAN communities, the strength-based focus of this survey tool makes it relevant to AIAN youth populations.

Reviewer #2 

In the title, I think the use of “youth” may not appropriate, as sample of this study focused merely on high school students with a median age of 15 years. We agree and we will change this to high school students.

“Psychometric evaluation” means the manuscript must present specific results of the survey and analyze their differences. But the “objective” of this study (line 112-114, page 5) was to validate the measurement properties of …. And on the other hand, as a research of reliability and validation, I don't think this study can reach that conclusion (more details given in Q3), so it would be more appropriate to call it a pilot study or a cross-sectional study. Psychometric implies measurement properties are evaluated to some extent. This can be done as well in cross-sectional studies as well through a variety of approaches for validity evidence, including internal validity via factorial validity for construct validity which we did. We revised the title to: Psychometric Evaluation of Protective Measures in Native STAND: A Multi-site Cross-Sectional Study of American Indian Alaska Native High School Students

The presentation of the manuscript is not rigorous enough, and some small mistakes are obvious.

 Thank you, we have addressed this concern. 

As far as known from the manuscript, sample number of this research was 1456 exactly, but line 44-45 in page 1, the author why said, “more than”? We removed the “more than” language. The original reason was we had some youth complete post data only but excluded them from this study because that was after the Native STAND curriculum, which we viewed as an intervention and wanted to evaluate properties only before/pre-intervention. The sample size at pre-Native STAND is indeed 1,456 participants which we have tried to make clear in the resubmission. 

Page 2, line 48, authors represented in the manuscript that the factor of negative self-esteem was two-item, so, factor loading of “0.65” was redundant Deleted thank you. 

I doubt about “resilience” you have mentioned in the keywords list, there is less content related to it. Removed from key word list.

Page 9, line 204, item ratings here were all included in the factor of “positive self-esteem”, it should be clearly defined and stated Revised thank you. 

Page 9, line 213, items of “history” and “native strengths” had the same rating of 3.8. This is not an error - the mean ratings were both 3.8 as initially reported. 

Some of the statistical procedures were not clearly explained or the methods were incorrectly used. Because of this issue, the conclusion that the measurement is a reliable and valid tool should be reconsidered or "tone-downed". Agreed and toned down conclusion and implications. 

As presented in figure 1. and figure 3. , parallel analysis of “self-esteem” and “social support” may indicate one more factor for each scale; these were also confirmed by the unsatisfactory total variance explanation (e.g., 53.9% of self-esteem and 59% of community of community safety). Please clarify. Figure 1 indicates two factors very clearly by the parallel analysis. We then conclude with a two-factor model: positive and negative self-esteem which has been found many times with the Rosenberg Self-Esteem scale before. So, there is no contradiction here between what we report and what parallel analysis suggests.

Preacher and MacCallum (2003) recommend parallel analysis (PA) as the way to choose the number of factors in an EFA – not total variance explained or in combination with PA. If we then choose more factors after the 2 that PA suggests, then PA disagrees with this and we are not using best practices to pick the number of factors / dimensionality as Preacher and MacCallum (2003) suggests. Reference: Preacher, K. J., & MacCallum, R. C. (2003). Repairing Tom Swift’s electric factor analysis machine. Understanding Statistics: Statistical Issues in Psychology, Education, and the Social Sciences, 2(1), 13–43.

Further, even other papers published in PLOS ONE itself report EFAs with such % total variance explained. This 2022 PLOS ONE reference: Fitriana, N., Hutagalung, F. D., Awang, Z., & Zaid, S. M. (2022). Happiness at work: A cross-cultural validation of happiness at work scale. PLOS ONE, 17(1), e0261617. https://doi.org/10.1371/journal.pone.0261617

Remarks (on bottom of page 6 of 16): “According to Hair, Black [17], total variance of 60% or even less than 60% is considered acceptable for social sciences.” So why is our presentation for PLOS ONE any different when following best practices utilizing PA as best way to choose the number of factors based on Preacher and MacCallum (2003)? When >1 factor we have reported appropriately as such.

Same response for Figure 3: PA found two factors – and we concluded two factors of social support with family and friends social support. We report separate factor loadings of each, their factor correlation, and separate internal consistency reliability estimates.

Page 6, Line 138-139, participants of this study were high school students, they could not represent “youth”, and we know that, there are fifty states in the U.S., how these 16 states sampled, it may need more clarification. Added and changed title to high school youth per Reviewer 1 comments. Thank you. 

Page 8, line 189-190, does any item was removed based on the EFA, and its procedure, I didn’t see it. Items were removed in the EFA procedure as previously described in the Data Analysis section of the original submission: An item was considered to load onto a factor if its factor loading was at least 0.40. Non-loading items and items loading on more than one factor were removed and the EFA re-ran. Thus, if an item loaded, it was not removed. 

Page 6, Line 139- 140: “Most of the sites were from rural communities and were from the western part of the United States.” Maybe the geographical distribution of the samples could be more elaborated in Table 1, for clarification We added the geographical distribution from the largest contributing states in a revised Table 1 for the resubmission.

Page 6, line 140, “a baseline survey” was done prior to participation, however, there is no description for the baseline. Please elaborate. This was an error, we revised it to read, youth completed the Native STAND survey. Thank you. 

Line 148-150, the study team added questions about many factors, but the Manu did not show their results, and if not, why you mention them here. Please clarify. These are not reported in this manuscript because they have been published elsewhere and are part of a larger study about the Native STAND curriculum. This manuscript focuses only on strength-based measures. We deleted that sentence and readers may access other publications on Native STAND to explore those questions in more detail. 

Table 1: The sample consisted of 1140 AI/AN, and 150 others. Regarding Race/Ethnicity data, if 1140+150=1290 students were included in this study, which does not equal No.= 87.8% and No.=12.2%? Please explain We revised Table 1 to include all 1,456 original participants who at least partially completed the Native STAND survey. 

The Race/Ethnicity question is “check all that apply” so that participants could have checked multiple races and Hispanic. We report the frequency and percent that answered only American Indian or Alaska Native and all Other non-White in the revised Table 1.

Part of “measures”

Questions generating methods of the four scale and the scoring rules were identical, I think, there is no need to repeat the same description four times. Thank you, we added a sentence these were the same for all and deleted them form the others. Thanks. 

Line 182, where is the “supplement 1.” I didn’t see it. It is included as a non-reviewed document and includes all of the questions and possible responses. We are including it again for your information and recommend that you contact PLOS if you cannot see the supplement. Thank you. 

Conclusion” of the study was just derived a “start point”, and very little valuable information can be drawn from the current statement for fellow researchers Revised and toned down and also added recommendations for future surveys at the individual, family, and community level- all will help fellow researchers.

---

## [Decision Letter · Decision Letter 1]

20 Apr 2022

PONE-D-21-24780R1Psychometric Evaluation of Protective Measures in Native STAND: A Multi-site Cross-Sectional Study of American Indian Alaska Native High School StudentsPLOS ONE

Dear Dr. Kelley,

Thank you for submitting your manuscript to PLOS ONE. After careful consideration, we feel that it has merit but does not fully meet PLOS ONE’s publication criteria as it currently stands. Therefore, we invite you to submit a revised version of the manuscript that addresses the points raised during the review process.

We look forward to receiving your revised manuscript.

Kind regards,

Richard Huan XU

Academic Editor

PLOS ONE

Reviewers' comments:

Reviewer's Responses to Questions

**Comments to the Author**

1. If the authors have adequately addressed your comments raised in a previous round of review and you feel that this manuscript is now acceptable for publication, you may indicate that here to bypass the “Comments to the Author” section, enter your conflict of interest statement in the “Confidential to Editor” section, and submit your "Accept" recommendation.

Reviewer #1: All comments have been addressed

Reviewer #2: All comments have been addressed

2. Is the manuscript technically sound, and do the data support the conclusions?

Reviewer #1: Yes

Reviewer #2: Partly

3. Has the statistical analysis been performed appropriately and rigorously? 

Reviewer #1: Yes

Reviewer #2: Yes

4. Have the authors made all data underlying the findings in their manuscript fully available?

Reviewer #1: No

Reviewer #2: Yes

5. Is the manuscript presented in an intelligible fashion and written in standard English?

Reviewer #1: Yes

Reviewer #2: Yes

6. Review Comments to the Author

Reviewer #1: The authors’ revisions and responses are appreciated. I have two minor suggestions as below.

1. For corrected item-total correlations, the authors could consider to specify which items are the ones with maximum and minimum correlation coefficients in the text.

2. In the response to one of the previous comments on self-esteem scale, the authors stated that “Previous research reports that the negative or reverse coded (RV) items were not as reliable as the positively coded items” in Rosenberg's self-esteem scale. Is it one of the reasons why authors excluded three RV items in current version of the scale? If yes, please consider to add this reason and the citation in Method section.

Reviewer #2: Thanks for inviting me again to review this interesting manuscript.

The authors made large modification to the manuscript, in response to reviewers’ comments. I think the paper still needs little modifications to fully meet the requirements for PLOS ONE to publication

Q1

Line 233, in the part of “discussion”, authors are still using the expression of “AI/AN youth”, which I think may generalize the results of the study. Authors should limit their findings to the sample they investigated, and consistent with the title

Q2

In the part of “Conclusion”, the authors stated too much about the value of this research for fellow researchers, I understand the necessity of these statements, but in fact, in the “conclusion”, you only need to tell the readers what were the most important finding through the “results” and “discussion” briefly. So, suggestion of research direction for fellow researchers can be moved to the part of "discussion", which I think was too thin.

In conclusion, I have no other comments on this manuscript, if the authors could improve the details, I think it will be valuable research for readers.

7. PLOS authors have the option to publish the peer review history of their article (what does this mean?). If published, this will include your full peer review and any attached files.

Reviewer #1: No

Reviewer #2: **Yes: **Lingming Zhou

---

## [Author Response · Author response to Decision Letter 1]

2 May 2022

May 2, 2022

Dear Editor,

We are pleased to resubmit our manuscript for consideration in PLOS One. Our responses to reviewer comments are located in the table below.

Please feel free to contact me if you have additional questions. 

Allyson Kelley, DrPH

kelleyallyson@gmail.com

Reviewer 1 Response 

Reviewer #1: The authors’ revisions and responses are appreciated. I have two minor suggestions as below.

1. For corrected item-total correlations, the authors could consider to specify which items are the ones with maximum and minimum correlation coefficients in the text.

 We appreciate this request and this information was in the paper under Table 2 on pages 7-8 for Self-esteem “Inter-item correlations ranged from 0.213 to 0.329 (average = 0.278) and corrected item-total correlations ranged from 0.361 to 0.628.”, and for Culture, Social support, and Community and Community Safety. This is also a column now in Table 2. Please let us know if you have additional questions. Thank you.

2. In the response to one of the previous comments on self-esteem scale, the authors stated that “Previous research reports that the negative or reverse coded (RV) items were not as reliable as the positively coded items” in Rosenberg's self-esteem scale. Is it one of the reasons why authors excluded three RV items in current version of the scale? If yes, please consider to add this reason and the citation in Method section. We appreciate this comment and added a citation to the methods section. The self-esteem scale we used included two RV coded items instead of three because of potential reliability issues with RV coded items reported by previous researchers

Marsh, H. W. (1996). Positive and negative global self-esteem: A substantively meaningful distinction or artifactors?. Journal of personality and social psychology, 70(4), 810.

The self-esteem scale was also modeled after the Native VOICES survey. We added appropriate citations in the methods section.

Reviewer 2 

Q1

Line 233, in the part of “discussion”, authors are still using the expression of “AI/AN youth”, which I think may generalize the results of the study. Authors should limit their findings to the sample they investigated, and consistent with the title We appreciate this feedback. We revised this sentence to be more consistent with the title and indicated results apply to those youth participating in the study. Thanks!

The revised title is, "Psychometric Evaluation of Protective Measures in Native STAND: A Multi-site Cross-Sectional Study of American Indian Alaska Native High School Students"

Q2

In the part of “Conclusion”, the authors stated too much about the value of this research for fellow researchers, I understand the necessity of these statements, but in fact, in the “conclusion”, you only need to tell the readers what were the most important finding through the “results” and “discussion” briefly. So, suggestion of research direction for fellow researchers can be moved to the part of "discussion", which I think was too thin.

In conclusion, I have no other comments on this manuscript, if the authors could improve the details, I think it will be valuable research for readers. We appreciate this feedback and edited the conclusion to highlight the most important finding. We moved the suggestions for future research to the discussion section as well. 

Thank you.

---

## [Editor Report · Decision Letter 2]

3 May 2022

Psychometric Evaluation of Protective Measures in Native STAND: A Multi-site Cross-Sectional Study of American Indian Alaska Native High School Students

PONE-D-21-24780R2

Dear Dr. Kelley,

We’re pleased to inform you that your manuscript has been judged scientifically suitable for publication and will be formally accepted for publication once it meets all outstanding technical requirements.

Kind regards,

Richard Huan XU

Academic Editor

PLOS ONE
---

## [Editor Report · Acceptance letter]

9 May 2022

PONE-D-21-24780R2 

Psychometric Evaluation of Protective Measures in Native STAND: A Multi-site Cross-Sectional Study of American Indian Alaska Native High School Students 

Dear Dr. Kelley:

I'm pleased to inform you that your manuscript has been deemed suitable for publication in PLOS ONE. Congratulations! Your manuscript is now with our production department. 

Kind regards, 

on behalf of

Dr. Richard Huan XU 

Academic Editor

PLOS ONE